# Influence of Host Sialic Acid Receptors Structure on the Host Specificity of Influenza Viruses

**DOI:** 10.3390/v14102141

**Published:** 2022-09-28

**Authors:** Chuankuo Zhao, Juan Pu

**Affiliations:** Key Laboratory for Prevention and Control of Avian Influenza and Other Major Poultry Diseases, Ministry of Agriculture and Rural Affairs, College of Veterinary Medicine, China Agricultural University, Beijing 100193, China

**Keywords:** influenza virus, sialic acids, host specificity

## Abstract

Influenza viruses need to use sialic acid receptors to invade host cells, and the α-2,3 and α-2,6 sialic acids glycosidic bonds linking the terminal sialic acids are generally considered to be the most important factors influencing the cross-species transmission of the influenza viruses. The development of methods to detect the binding of influenza virus HA proteins to sialic acid receptors, as well as the development of glycobiological techniques, has led to a richer understanding of the structure of the sialylated glycan in influenza virus hosts. It was found that, in addition to the sialic acid glycosidic bond, sialic acid variants, length of the sialylated glycan, Gal-GlcNAc-linked glycosidic bond within the sialylated glycan, and sulfation/fucosylation of the GlcNAc within the sialylated glycan all affect the binding properties of influenza viruses to the sialic acid receptors, thus indirectly affecting the host specificity of influenza viruses. This paper will review the sialic acid variants, internal structural differences of sialylated glycan molecules that affect the host specificity of influenza viruses, and distribution characteristics of sialic acid receptors in influenza virus hosts, in order to provide a more reliable theoretical basis for the in-depth investigation of cross-species transmission of influenza viruses and the development of new antiviral drugs.

## 1. Introduction

Influenza viruses belong to the family *Orthomyxoviridae* and comprise a family of four distinct viruses: influenza A, B, C, and D. The natural host of influenza A viruses (H1–H16) is wild waterfowl, but they can also infect a wide range of poultry and mammals, such as chickens, ducks, horses, pigs, dogs, seals, and minks. Influenza A viruses are one of the zoonosis and have caused four human influenza pandemics to date. The most serious was the 1918 influenza pandemic, followed by the 1957 Asian and 1968 Hong Kong flus, both of which formed world pandemics; in the 21st century, a new reassortment of the H1N1 subtype of swine flu virus caused a global pandemic in 2009, which has now evolved into a seasonal human flu virus and caused a large number of severe patients and even deaths [1,2].

Animal influenza A viruses circulating in livestock and poultry have always been considered the greatest potential threat to humans. Various novel influenza viruses reassorted from swine, avian, and human influenza viruses in different periods that are widely prevalent in pig herds, which often spread across species and causes human infection or death. In addition, H5, H7, and H9 subtypes are the most common influenza virus types in poultry flocks. Human infection cases have recently occurred frequently and are listed by the World Health Organization (WHO) as important viruses with the potential to cause the next human pandemic [3,4,5,6,7]. Influenza A virus has a broad host spectrum, and the cross-species transmission of animal influenza to humans is becoming more and more frequent; some animal influenza viruses are considered to have the potential to cause a human pandemic, so the relevant research on the key factors of influenza virus cross-species transmission remains critical.

The binding of influenza virus hemagglutinin (HA) proteins to host cell sialic acid receptors is the first step in the invasion of host cells by influenza viruses and considered to be the first barrier to cross-species transmission. Thus, sialic acids distribution characteristics in host cell surface influence host specificity of influenza virus [8]. With recent developments in glycobiological techniques, researchers have gained new insights into the interaction of influenza viruses with different sialylated glycan molecules and the distribution characteristics of sialic acid receptors in the hosts [9,10]. This paper will review the effects of sialic acids variants, internal structural differences of sialylated glycan molecules on the host range of influenza viruses, and distribution characteristics of sialic acid receptors in hosts, in order to provide a more reliable theoretical basis for clarifying the cross-species transmission mechanism of influenza viruses and the development of antiviral drugs.

## 2. Influenza Virus Sialic Acid Receptors

The surface of host cells is covered by complexes, such as polysaccharides, which are one of the necessary substances for cells to carry out their vital activities and play an important role in many physiological and pathological processes. As well, the polysaccharides exposed on the extracellular surface play an important role in the attachment of influenza viruses. Many types of polysaccharides, N-linked glycans, O-linked glycans, glycolipids, etc., are present on the cell surface. Influenza virus infection is initiated by binding of HA protein on the viral envelope to epithelial cell surface-exposed sialic acid receptors in enteric tract and respiratory tract of the host (Figure 1). Sialic acid is a nine-carbon monosaccharide. N-acetylneuraminic acid (Neu5Ac) and N-glycolylneuraminic acid (Neu5Gc) are two of the most common forms of sialic acids on the host cell surface. Neu5Ac is regulated by the the enzyme CMP-Neu5Ac hydroxylase (CAMH) to undergo hydroxylation to form Neu5Gc [11,12]. Influenza viruses from different host sources have different binding abilities to these two sialic acids forms. However, it is the specific structural features resulting from differences, in the way in which the terminal sialic acids linked to the sub-terminal galactose, that have a greater effect on the specificity of influenza virus receptors binding. The C2-position carbon atom of sialic acids can be linked to the C3- or C6-position carbogen of the secondary terminal galactose via an α-2,3 sialyltransferase (ST3Gal) or α-2,6 sialyltransferase (ST6Gal)-catalyzed α-2,3 or α-2,6 glycosidic bond, and the effect of different sialic acids of α-2,3 or α-2,6 glycosidic bonds on the binding properties of the influenza virus receptors is currently commonly considered as a major factor limiting the cross-species transmission of influenza viruses [8]. In addition, the detection of influenza virus–sialic acid receptor interactions is becoming more and more abundant and has evolved from the initial erythrocyte adsorption assay to solid-phase direct binding assay, glycan microarrays, surface plasmon resonance (SPR), and X-ray [10,13]. Some of the sialic acid receptors used in current studies are chemically synthesized according to the structure of sialic acids in the American Functional Glycomics Group Consortium for Functional Glycomics (CFG, http://www.functionalglycomics.org/, accessed on 31 August 2022), while others are chemically synthesized according to the distribution characteristics of sialic acids in the human respiratory tract [14,15,16]. It has been found that the length of sialylated glycan molecules, glycosidic bonds between the internal monosaccharides, and sulfation/fucosylation of the internal monosaccharides also have an effect on the receptor binding properties of the influenza virus. The more studied sialic acids analogues that can affect the binding properties of influenza virus receptors are sialylated glycan molecules with extended poly-N-Acetyl-D-lactosamine (poly-LacNAc), sialylated glycan molecules with different glycosidic linkages of the penultimate galactose (Gal), antepenultimate N-acetylglucosamine (GlcNAc), and sialylated glycan molecules with sulfation/fucosylation of the antepenultimate monosaccharide (Figure 1). Therefore, the host specificity of these more complex sialic acid receptors for influenza viruses needs more attention.

## 3. Effect of the Sialic Acids Variant Neu5Gc on the Host Specificity of Influenza Viruses

Neu5Ac and Neu5Gc are the two most predominant forms of sialic acids in animal cells. Most influenza viruses readily bind to Neu5Ac and, in recent years, it has been found that influenza viruses can also invade host cells by binding to Neu5Gc. Studies have shown that Neu5Gc is not present in the respiratory tract of humans and ferrets, but is present in the respiratory tract of horses and pigs [17]. The H7N7 subtype of equine influenza virus binds specifically to Neu5Gc, and the H3N8 subtype of equine influenza virus also binds preferentially to Neu5Gc, but the H3N8 subtype prevalent in dogs binds preferentially to Neu5Ac. It is noteworthy that cross-species studies of avian H3N8 infection in humans have recently been reported in China; how this avian H3N8 differs from other host sources of H3N8 needs to be further investigated (Figure 2) [18,19,20,21]. The human influenza A/Memphis/1/71 (H3N2) T155Y mutation maintains the dual receptors binding properties of Neu5Ac and Neu5Gc, with the exception of equine influenza virus, which is readily compatible with Neu5Gc. The H5N1 A/Vietnam/1203/04 Y161A mutant interacts with Neu5Gc, and the virus replicates well in cell cultures [17,18]. In addition, to verify the effect of Neu5Gc on virus evolution, pandemic H1N1 human influenza virus and seasonal H3N2 human influenza virus were passaged on the surface of respiratory tissues in a mouse model with Neu5Gc, and only a few mutant sites were found, indicating that the mouse model with Neu5Gc did not show strong selective evolution in influenza virus infection, probably due to the host-dependent effect of Neu5Gc on virus evolution [22]. It is also noteworthy that Neu5Gc is also present in pigs, but whether the currently prevalent swine influenza virus can bind Neu5Gc needs to be urgently addressed [17].

## 4. Effect of Sialic Acids Glycosidic Bonds on the Host Specificity of Influenza Viruses

It is commonly believed that human influenza viruses preferentially bind α-2,6 sialic acids, while avian influenza viruses preferentially bind α-2,3 sialic acids, and that sialic acids glycosidic bonds affect cross-species transmission of influenza viruses. The influenza virus receptors binding site is located at the distal membrane end of the HA molecule in a shallow, concave pocket consisting of a series of conserved amino acids at the base, including Y98, W153, H183, and Y195 (H3 code), flanked by secondary structural elements, including the 130-loop, 150-loop, 190-helix, and 220-loop [23]. Among all the HA -receptor complex crystal structures that have been resolved, most of the α-2,3 sialic acid receptors take a trans conformation, and the α-2,6 sialic acid receptors usually take a cis conformation, in order to adapt to the different conformations of α-2,3 sialic acid and α-2,6 sialic acid receptors. The amino acids at the HA receptor binding site also need to change accordingly [24,25]. Therefore, the change in receptor binding specificity caused by amino acid variations at the HA receptor binding site is the molecular basis for the occurrence of cross-species transmission of influenza viruses.

For H1 subtype viruses, amino acids at positions 190 and 225 play a key role in the determination of receptor binding specificity. The H1 subtype avian influenza virus is mainly E190 and G225, which can bind to α-2,3 sialic acids. The H1 subtype human influenza virus is D190 and D225, which binds only α-2,6 sialic acids [23,26,27,28]. It was found, by structural analysis, that the H1 subtype HA proteins form hydrogen-bonded interactions with the α-2,6 sialic acids via amino acid residues D190 and D225, and the salt-bridge interaction between D225 and K222 reduces the flexibility of the 220-loop, which is detrimental to the binding of the α-2,3 sialic acids. For H2 and H3 subtypes of avian influenza viruses, the Q226 L and G228S mutations on the HA protein (encoded by H3) are key to altering receptor binding properties and enabling cross-species transmission [1]. Avian influenza virus HA containing Q226 and G228 preferentially binds α-2,3 sialic acids, whereas human influenza virus HA containing L226, and S228 specifically binds α-2,6 sialic acids [23,29,30]. The hydrophobic environment provided by L226 favors the binding of α-2,6 sialic acids to the detriment of α-2,3 sialic acids, while the formation of a hydrogen bond between S228 and sialic acid increases the affinity of HA for α-2,6 sialic acids.

Most people infected with H5N1 subtype influenza viruses have a history of avian exposure, and there is insufficient evidence that H5 subtype influenza viruses have the ability to bind the α-2,6 sialic acid receptors. However, in the laboratory, researchers have modified H5N1 mutant viruses that specifically bind the α-2,6 sialic acid receptors and are airborne between mammals. Kawaoka et al. [31] reported that mutations N158D, N224K, Q226L, and T318I were introduced into the HA protein of influenza virus A/Vietnam/1203/2004 (H5N1), gaining the ability to bind α-2,6 sialic acids and enabling airborne transmission in the model animal ferret. Structural studies have shown that the hydrophobic environment created by residue L226 allows the mutant H5 subtype HA protein to bind α-2,6 sialic acids, to the detriment of the α-2,3 sialic acid receptors, consistent with the principle that the H2 and H3 subtypes of human influenza virus HA proteins selectively bind the α-2,6 sialic acid receptors; in addition, the N158D or T160A mutations result in the deletion of an N-glycosylation site, at position 158, near the HA receptor binding site, thereby increasing the affinity for the α-2,6 sialic acid receptors [32,33,34,35].

In the outbreak of the H7N9 virus infection in humans in 2013, studies on the receptor-binding properties of H7N9 virus showed that H7N9 virus still retains the ability to bind to α-2,3 sialic acid receptors, but most strains have acquired some ability to bind α-2,6 sialic acid receptors [36]. The HA proteins of A/Shanghai/1/2013 (H7N9) and A/Anhui/1/2013 (H7N9) have only eight amino acid differences in the primary sequence, of which four amino acid differences are in the receptor binding site. S138A, G186V, T221P, and Q226L together created a hydrophobic environment, making the region near the 220 loop of the receptor binding site of A/Anhui/1/2013 more hydrophobic than that of A/Shanghai/1/2013, thus making it easier to bind to α-2,6 sialic acids. Even after the L226Q mutation, the other three hydrophobic amino acids still provided a sufficient hydrophobic environment to maintain the binding of the α-2,6 sialic acids. This result suggests that, unlike the H5N1 virus, the Q226L amino acid mutation is not the only critical factor in the ability of the H7N9 virus HA to acquire α-2,6 sialic acid receptors binding, but that other related amino acids in the receptor binding site are also critical [36,37].

Notably, the H9N2 avian influenza viruses isolated in China in the late 20th century showed a preference for binding to the α-2,6 sialic acid receptors [38]. Subsequently, the ongoing surveillance of H9N2 subtype avian influenza viruses has revealed that some currently prevalent strains have evolved to bind only the α-2,6 sialic acid receptors, but not the α-2,3 sialic acid receptors [38,39]. Preliminary progress has been made on the mechanism of α-2,6 sialic acid receptors preference of H9N2 subtype avian influenza viruses, with the amino acids at positions 190, 226, and 227 of the HA gene playing a major role in receptor preference [40,41]. In addition, it has been shown that the combined mutation Q226L and Q227M enhances the binding properties of α-2,6 sialic acid receptors [42]. However, amino acids L and Q at position 226, A, V, and T at position 190, and Q and M at position 227 have been isolated from H9N2 viruses in nature. Random combinations of these three amino acid sites exist, and the different combinations have different receptor binding properties for H9N2 viruses [41,43]. Apart from this, the sporadically infected human H10 and H6 subtypes of avian influenza viruses still retain their alpha-2,3 sialic acids receptor binding properties [44,45,46,47].

In summary, as avian influenza viruses are now infecting humans across species with increasing frequency, the receptor-binding properties of H9N2 influenza viruses appear to contradict the common perception that avian influenza viruses recognize α-2,3 sialic acid receptors and human influenza viruses recognize α-2,6 sialic acid receptors. Furthermore, the mechanism behind why H9N2 subtype avian influenza viruses that have completely shifted to α-2,6 sialic acid receptors binding properties can still replicate and spread well in poultry has not been explained. In addition to the sialic acids glycosidic bond, the internal structure of the sialylated molecules also has a significant impact on the receptor binding properties of influenza viruses. The recent finding that H9N2 avian influenza viruses readily bind to sulphated modified α-2,3 sialic acid receptors also provides a new perspective to elucidate how well H9N2 subtype avian influenza viruses with α-2,6 sialic acid receptors binding properties replicate in poultry.

## 5. Influence of the Internal Structure of the Sialylated Glycan Molecules on the Host Specificity of Influenza Viruses

### 5.1. Influence of Sialylated Glycan Length on the Host Specificity of Influenza Viruses

As the human seasonal influenza H3N2 virus evolved, it gradually lost its ability to agglutinate chicken red blood cells and was found to have a reduced ability to recognize α-2,6 sialic acid receptors by conventional receptor binding characteristics assays [30]. However, subsequent studies found that human seasonal influenza H3N2 influenza viruses showed progressively greater recognition of α-2,6 long-chain sialic acid receptors with LacNAc repeat sequences [14]. It was suggested that the strong preference for the α-2,6 long-chain sialic acid receptors may be due to the potential for increased avidity by binding to two of the monomers of the HA trimer [14]. The increased specificity of the H3N2 subtype of human influenza isolates with α-2,6 long-chain sialic acids sugars in recent years is mainly due to the process of “avidity maturation”, where the H3 protein is evolving into a binding mode through multivalent interactions with sialic acid receptors, while losing its specificity with shorter linear receptors [14]. Additional studies have shown that the 2009 pandemic H1N1 influenza virus can also bindα-2,6 long-chain sialic acids, and that the H7N9 subtype of avian influenza virus enhances binding to long-chain α-2,6 sialic acids through mutations in the HA protein K193T [48]. When investigating the specific physiological anatomical sites of replication of airborne influenza viruses, it was found that the soft palate and nasal respiratory epithelium of ferrets were the main physiological anatomical sites of virus replication, while the sialic acid receptors located in the soft palate of ferrets were mostlyα-2,6 long-chain sialic acid receptors. Thus, influenza virus binding to long-chain α-2,6 sialic acid receptors may be necessary for the ability of influenza viruses to transmit [49,50].

### 5.2. Influence of Gal-GlcNAc/GalNAc-Linked Glycosidic Bonds on the Host Specificity of Influenza Viruses

The differences in the glycosidic bonds linking the penultimate Gal and third monosaccharide GlcNAc/GalNAc at the terminus resulted in Neu5Ac 2-3 Galβ1-4 GlcNAc (3′SLN), Neu5Ac 2-3 Gal β1-3 GlcNAc(SLe^c^), and Neu5Ac 2-3 Gal β1-3 GalNAc (3′SiaTF). The differences in the glycosidic bonds linking Gal-GlcNAc/GalNAc affected the adaptation of avian influenza viruses in ducks and chickens. Although avian influenza viruses infecting both chickens and ducks were able to bind α-2,3 sialic acid receptors, avian influenza viruses infecting ducks were less likely to infect chickens directly under experimental conditions; conversely, avian influenza viruses infecting chickens were also less likely to infect ducks [51,52,53]. Studies have shown that avian influenza viruses isolated from terrestrial birds are more likely to bind 3′SLN, and the avian influenza viruses of duck origin preferentially bind SLe^c^ and 3′SiaTF [54,55,56,57,58,59].Thus, monosaccharide-linked glycosidic bonds within sialylated glycan molecules affect the host specificity of influenza viruses in avian species.

### 5.3. Influence of the Sulfation/Fucosylation of the GlcNAc within Sialylated Glycan Molecules on Host Specificity of Influenza Viruses

The fucosylation occurs on the sialylated glycan molecules internal monosaccharide GlcNAc and affects the binding properties of influenza viruses, similarly to the Gal-GlcNAc-linked glycosidic bond, with terrestrial avian isolates more likely to bind Neu5Acα2-3Galβ1-4(Fucα1-3))GlcNAc (SiaLe^x^) and Neu5Acα2-3Galβ1-4(Fucα1-3)-(6-HSO_3_)GlcNAc (6-suSiaLe^x^) [54,56]. The chicken-derived H5N1 subtype of influenza virus has been identified to bind SiaLe^x^ since the early 21st century, and the associated crystal structure has been resolved [60]. Hiono et al. [61] reported that the low pathogenic H5N2 subtype of avian influenza viruses isolated from chickens preferentially bound to SiaLe^x^. The altered receptor binding specificity of the 2.3.4.4 clade H5 protein may have contributed to the emergence of the H5Nx virus, as compared to the 2.3.4 clade H5 protein, in which the K222Q and S227R mutations enhance binding to SiaLe^x^ [62]. In addition, the W222L mutation in the HA protein of the horse-derived A(H3N8) virus increased the affinity of the viral HA protein to bind specifically to the SiaLe^x^ receptors, thereby facilitating the adaptation to dogs [20]. Some seasonal influenza viruses are also able to bind to fucosylation of the GlcNAc within the sialylated glycan molecules [63].

The sulphated modification occurs on the internal monosaccharide GlcNAc of sialylated glycan molecules also affects the binding properties of influenza viruses. The terrestrial H5/H7 subtype avian influenza virus that prefer to bind Neu5Acα2-3Galβ1-4(6-HSO_3_)GlcNAc (3′SLN-6-O-sulfate) [56,59]. It is worth noting that the majority of avian influenza viruses circulating in humans preferentially bind 3′SLN-6-O-sulfate. The H7N9 subtype of avian influenza virus that broke out in 2013 has dual α2-3 and α2-6 sialic acids receptor binding properties and binds 3′SLN-6-O-sulfate [64]. Subsequent reports of sporadic human infections with avian influenza subtype H10 viruses still maintain alpha-2,3 sialic acids receptor binding properties and are able to bind to 3′SLN-6-O-sulfate [45]. Most studies have shown that the H9N2 subtype of avian influenza virus is currently able to bind to the α-2,6 sialic acids, but there are also reports in the literature that the H9N2 subtype of avian influenza virus is currently able to bind to 3′SLN-6-O-sulfate [41]. This suggests that H9N2 subtype avian influenza virus has the potential to replicate in birds via sulphated α-2,3 sialic acids. Mutations in the HA protein E190V of H6N1 subtype avian influenza virus are essential for recognition of 3′SLN-6-O-sulfate [51]. It has also been shown that the human influenza virus H1N1 enhanced binding to 3′SLN-6-O-sulfate by passaging in chicken embryos, and it has enhanced replication in chicken embryos [65]. The virus receptor binding characteristics currently performed in the laboratory may have been passed through chicken embryos, which could potentially lead to enhanced binding to the sulphated 2,3 sialic acid receptors.

## 6. Characteristics of the Distribution of Sialylated Glycan Molecules in the Host

The previous article has reviewed the binding of influenza virus to different sialylated glycan molecules, and the distribution characteristics of these different sialylated glycan molecules in influenza virus hosts have also made breakthroughs with the development of glycobiology technology. Lectin immunohistochemistry is a common method for studying the distribution and expression of sialylated glycan molecules in tissues and cells, and Maackia amurensis agglutinin (MAA) and Sambucus nigra agglutinin (SNA) recognise α-2,3 sialic acid and α-2,6 sialic acid receptors, respectively. Among them, MAA is divided into MAA-1 and MAA-2. MAA-1 mainly recognizes Neu5Acα2-3Galβ1-4GlcNAc, while MAA-2 mainly identifies Neu5Acα2-3Galβ1-3GlcNAc; however, MAA-1 and MAA-2 also recognize structures with sulfated modified sugars [9,66]. In addition, some specific sialic acids structures are identified by immunohistochemistry using specific antibodies [53]. This method could only initially identify the distribution of α-2,3 sialic acid and α-2,6 sialic acid receptors. Subsequently, mass spectrometry was applied to characterize the distribution of salivary acids in hosts, especially in humans, pigs, and ferrets, which not only identified the distribution of α-2,3 sialic acid and α-2,6 sialic acid receptors, but also provided more detailed information on the structure of the internal sialic acids glycan chain, including the number of “antennae”, sulfation/fucosylation of internal monosaccharide, and LacNAc repeat sequences, etc. [9,67]. The application of mass spectrometry to the characterization of sialic acids distribution in influenza virus hosts has greatly enriched the information related to the structure of sialylated glycan molecules, but most of them are currently applied to the characterization of sialic acids distribution in mammalian hosts, while only chickens are currently identified by mass spectrometry in avian species (Figure 3) [67,68,69,70].

### 6.1. Characteristics of the Distribution of Sialic Acid Receptors in the Mammalian Host

Unlike other mammals, in normal humans, only Neu5Ac is present, and Neu5Gc cannot be detected; however, Neu5Gc is present in some human cancer cells [8,17,71]. Both the α-2,3 sialic acid and α-2,6 sialic acid receptors were detected on ciliated epithelial cells and mucus-secreting cells in the human nasal cavity, but were weakly bound to lectins. In the bronchi, the α-2,3 sialic acid and α-2,6 sialic acid receptors were heterogeneously distributed, with no clear distinction between ciliated and non-ciliated cells, but epithelial cells contained more α-2,3 sialic acid receptors. MAA-2 staining showed an abundant distribution of α-2,3 sialic acid receptors on type II alveolar cells [72,73,74]. However, due to differences in lectin suppliers, different studies have shown slightly different proportions of α-2,3 sialic acid and α-2,6 sialic acid receptors. It is now generally accepted that the human upper respiratory tract has a higher distribution of α-2,6 sialic acid receptors than the lower respiratory tract, and it has also been found that children have more α-2,3 sialic acid receptors in their lungs. Mass spectrometry of sialic acids distribution in the human bronchi and lungs also demonstrated the presence of α-2,3 and α-2,6 sialic acid receptors in both organs, which are mostly Sialylated Glycan Molecules with multiple “antennae” and sialic acids glycoconjugates containing a high number of LacNAc repeats and internal glycoconjugates with fucosylation [16,68,75]. Information on Sialylated Glycan Molecules in the upper respiratory tract is unknown because of the difficulty of obtaining N-linked sugars in the nasal cavity.

Expression of both α-2,3 sialic acid and α-2,6 sialic acid receptors in the respiratory tract of pigs [3,76]. In pigs, α-2,6 sialic acid receptors were predominantly distributed in the epithelial cells of the upper respiratory tract cilia, while the proportion of α-2,3 sialic acid and α-2,6 sialic acid receptors did not differ significantly in the pig lung. The lamina propria of the respiratory mucosa was dominated by α-2,3 sialic acid receptors, whereas α-2,6 sialic acid receptors were confined to mucus/plasma glands [77,78]. In contrast, mass spectrometry revealed a large number of α-2,6 sialic acid receptors in porcine trachea and lung, which are mostly sialic acids glycan chains with multiple “antennae”, and sialic acids contains more LacNAc repeat sequences. [69,76,79,80].

Ferrets are widely used as animal models of influenza virus pathogenicity and transmission because the distribution of sialic acid receptors in their respiratory tract is similar to that of humans [50,81,82,83]. A large number of α-2,6 sialic acid receptors are found in the tracheal and bronchial ciliated cells and submucosal glands in ferrets, while α-2,3 sialic acid receptors are found in the lamina propria; both receptors are expressed in alveoli, but α-2,6 sialic acid receptors are predominant [9,69,84]. Immunohistochemical with lectins and mass spectrometric studies have shown that both α-2,3 and α-2,6 sialic acid receptors are distributed in ferret respiratory organs, with the latter being more prevalent. Similar to the distribution of sialic acids in human respiratory tract tissues, sialic acids in ferrets are multiple “antennae”, and sialylated glycan molecules contain more LacNAc repeats and internal monosaccharides with fucosylation [67]. A variety of long-chain α-2,6 sialic acid receptors containing more LacNAc repeats are distributed in the soft palate of ferrets, and this physiological and anatomical site may be an important site for the production of transmissible virions [49,84]. Similar to human airway tissue, NeuGc does not exist in ferrets, but a new sialic acids structure Sda blood group antigen exists in ferrets [17,67].

The BALB/C mouse is currently the most commonly used animal model for studying influenza viruses. Studies have shown that α-2,3 sialic acid and α-2,6 sialic acid receptors are evenly distributed in the nasal base and connective tissue of the upper respiratory tract of mice. In the trachea and bronchi, α-2,3 sialic acid receptors were focally distributed, whereas α-2,6 sialic acid receptors were widely distributed, in the ciliated cells of the trachea and ciliated and non-ciliated cells of the bronchi [72,85]. In addition, both receptors were distributed in the lung, cerebellum, spleen, liver, kidney, and other organs or tissues of BALB/c mice [86]. However, due to differences in lectin suppliers, results from different studies have shown slightly different proportions of α-2,3 sialic acid and α-2,6 sialic acid receptors, and it is now generally accepted that mice have more α-2,3 sialic acid receptors distributed in their lungs, suggesting that mice are more suitable as an animal model for assessing the pathogenicity of avian influenza. Compared to human respiratory tissues, mice are similar to pigs in the presence of NeuGc [17].

The guinea pig has unique advantages over mouse and ferret animal models. It can be used as a model for assessing both the pathogenicity and transmission of avian influenza viruses, as well as inexpensive and easy to handle. The guinea pig contains both α-2,3 sialic acid and alpha-2,6 sialic acid receptors. The nasal and tracheal receptors are comparable in both, but the lungs and other organs are predominantly α-2,3 sialic acid receptors, and the guinea pig also has NeuGc [17,87].

In addition, there are other mammalian hosts capable of infecting influenza virus. Most of them were identified by lectin staining and immunohistochemical. The horse trachea was mainly NeuGc, most of which were α-2,3 NeuGc. SiaLe^x^ is absent in vivo [20,88,89,90]. While the canine respiratory airways and nasal tissues contain mainly α-2,3 sialic acid receptors and the specific structure SiaLe^x^, it is unclear whether NeuGc is present in the respiratory tract [9,17,20].

### 6.2. Characteristics of the Distribution of Sialic Acid Receptors in the Avian Host

Chickens are the main host of avian influenza viruses and can be infected with many subtypes of avian influenza viruses. In chickens, both α-2,3 sialic acid and α-2,6 sialic acid receptors are present, but the abundance of their distribution in tissues are different [9,66,91]. The different results of lectin immunohistochemistry methods may be due to different lectin manufacturers or different breeds and ages of chickens, but it is certain that both α-2,3 sialic acid and α-2,6 sialic acid receptors are present in chickens. The chicken is the only avian species that has been characterised by mass spectrometry for the distribution of sialic acids in the host, and studies have shown the presence of α-2,3 sialic acid and α-2,6 sialic acid receptors, which are multiple ‘antennae’ sialic acids glycoconjugates. Sulphated modified sialic acids and LacNAc repeat sequence sialic acids are present in the lungs, and specific sialic acids with fucosylation is present in the trachea, e.g., SiaLe^x^ [70].

Ducks are usually the vector for the transmission of avian influenza viruses from terrestrial birds to wild birds. The distribution of sialic acids in ducks is characterised by the presence of α-2,3 sialic acid and α-2,6 sialic acid receptors throughout the respiratory organs, with α-2,3 sialic acid receptors predominating and α-2,3 sialic acid receptors widely present in the intestinal tract of ducks [53,91].

α-2,3 sialic acid and α-2,6 sialic acid receptors are distributed in different abundances in birds, but both types of receptors are distributed in the avian respiratory tract, so some birds may also be sites of recombination of human influenza and avian influenza viruses [66,91,92]. As there are also differences in the distribution characteristics of sialic acids in different avian species, which can also lead to barriers to cross-species transmission of avian influenza viruses in birds, there are relatively few relevant studies reported, and they have not yet attracted widespread attention.

## 7. Summary and Future Directions

In addition to the α-2,3 and α-2,6 sialic acids glycosidic bonds that affect the host range of influenza viruses, the variants and internal structure of sialic acid receptors also affect the receptor binding properties of influenza viruses. While current research has shed new light on the receptor-binding properties of influenza viruses, there are still a number of questions that need to be addressed. For example, (1) the surface sialic acids receptor characteristics of mammalian host cells of influenza viruses are currently well-studied, while there are still many gaps in the distribution characteristics of sialic acid receptors in avian hosts. Although studies on the distribution characteristics of sialic acids in mammalian have been more in-depth than that in the avian hosts, more detailed spatial distribution studies on the respiratory tissues of influenza virus mammalian cells, combined with single-cell sequencing, mass spectrometry, and integrated multi-omics, are needed for a comprehensive understanding. (2) Current studies on the binding properties of influenza virus receptors using glycan microarrays have shown that influenza viruses can bind different types of sugars, but there are significant differences between the glycans currently used in glycan microarrays and types of sialic acids actually present in the host. Therefore, similar methods, such as shotgun glycomics, can be further used to explore the in vivo sialylated glycan binding characteristics of influenza viruses in the host. (3) The role of SiaLe^x^ in the cross-species transmission of influenza viruses needs to be further explored, as SiaLe^x^ can enhance the replication of influenza viruses from multiple host sources. (4) With the development of glycobiological techniques, the scientific community has gained new insights into influenza viruses and is gradually changing common concepts. In addition to sialic acid receptors, influenza viruses can also invade cells, with the help of non-sialic acid receptors, such as lectins [93,94,95,96,97,98,99,100,101,102], and there are differences in the surface forms of glycoproteins between the avian influenza and human influenza viruses, which are also important factors affecting the cross-species transmission of influenza viruses. In conclusion, the invasion of cells by influenza A viruses is an extremely complex process, and we still need to find patterns through exploration, in order to control avian influenza viruses.

## Figures and Tables

**Figure 1 viruses-14-02141-f001:**
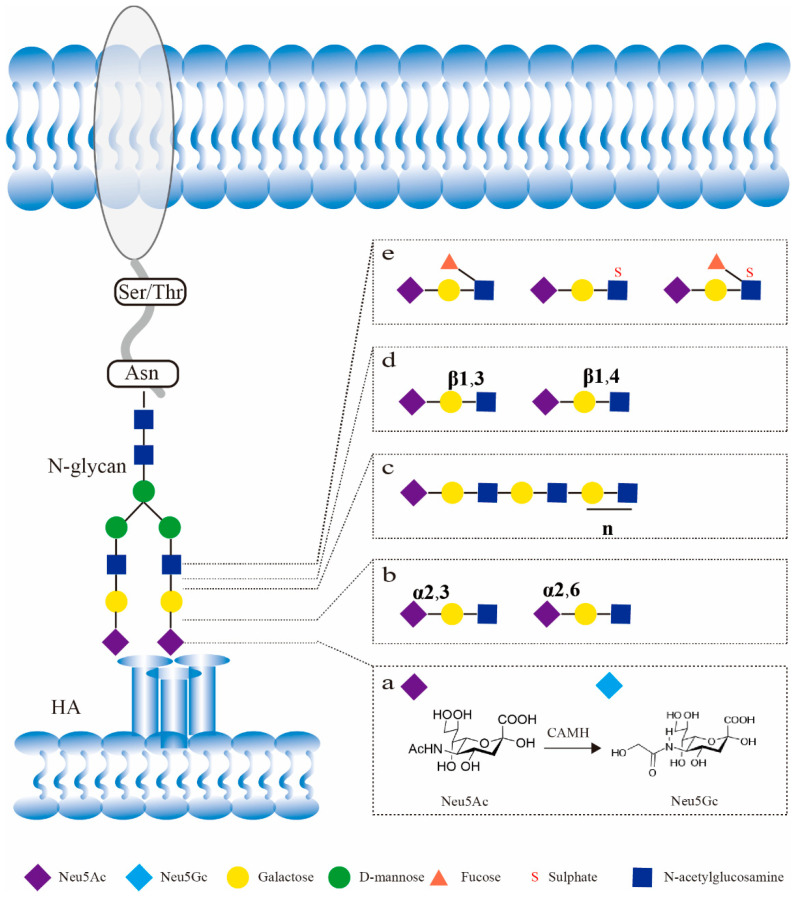
Influenza virus binds different sialylated molecules. (**a**): Neu5Ac and Neu5Gc are two forms of sialic acids. (**b**): Terminal sialic acids can be linked to secondary terminal galactose via α-2,3 and α-2,6 glycosidic bonds to form α-2,3 sialic acids and α-2,6 sialic acids. (**c**): Sialylated glycan molecules with extended poly-LacNAc. (**d**): Gal-GlcNAc can be linked by β-1,3 or β-1,4 glycosidic bonds to form different sialylated molecules. (**e**): GlcNAc can be fucosylated or sulphated to form different sialylated molecules.

**Figure 2 viruses-14-02141-f002:**
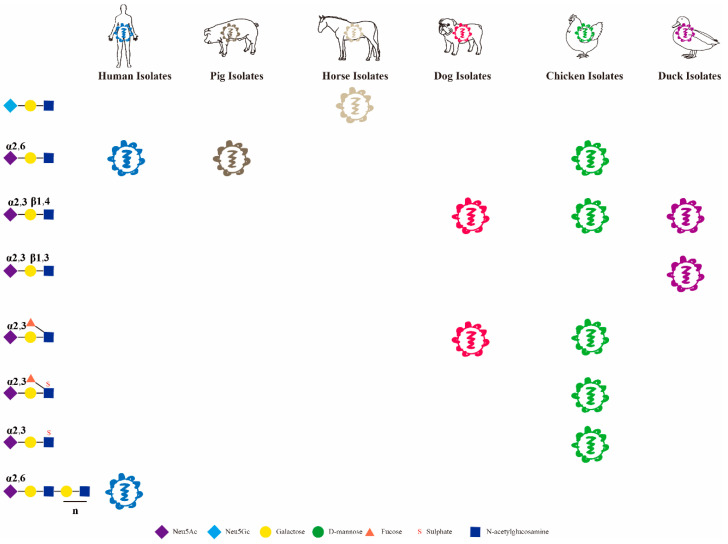
Preferential binding of influenza viruses of different host origins to sialic acids.

**Figure 3 viruses-14-02141-f003:**
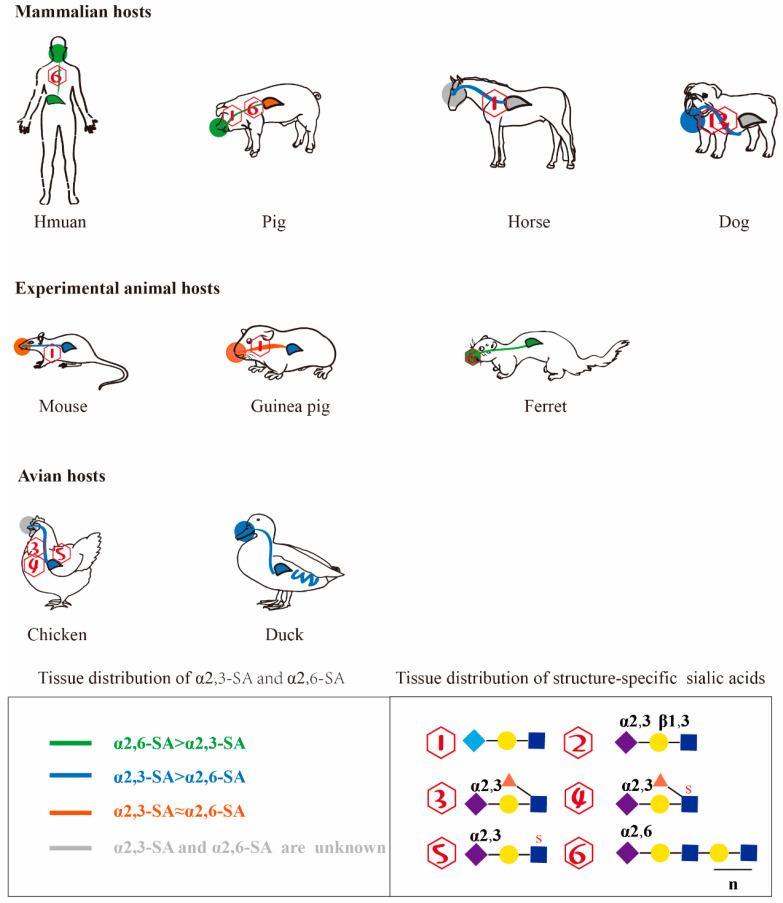
Characteristics of the distribution of sialylated glycan molecules in respiratory tract tissues of mammalian, experimental animal, and avian hosts.

## Data Availability

Not applicable.

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
