# Peer review of "Influence of Host Sialic Acid Receptors Structure on the Host Specificity of Influenza Viruses"

_viruses, 2022, doi:10.3390/v14102141_

Round 1

Reviewer 1 Report

This review paper entitled Influence of host sialic acid receptor structure on the host specificity of influenza viruses, by Chuankuo Zhao and Juan Pu, reviews the sialic acid variants, the internal structural differences of sialic acid that affect the host specificity of influenza viruses and the distribution characteristics of sialic acid receptors in influenza virus hosts.

The review provides a good summary for the role of sialic acid variants in the infection with influenza virus and provides insight into cross-species transmission of influenza viruses. This is important for the development of new antiviral drugs and pandemic preparedness.

The review is well written, however, some issues need to be improved.

Line 9:                   Should be Abstract

Line 58:                 Please explain in Section 2 what kind of cells /tissues bearing sialic receptors represent target for influenza virus and where these tissues are located.

Figure 2:               Upper row Indicated two Horse isolates and two Gog isolates. Did authors mean Human and Pig isolates? Please check Figure 2 is correct.

Author Response

Thank you for the helpful comments. Based on the comments and suggestions, we have made careful modifications to the original manuscript. Details of our point-by-point responses are in the attachment,and all revisions made to the manuscript have been marked up using the“Track Changes” function.

Reviewer 2 Report

The topic of this article is interesting, although not necessarily new. Most of the data have been reported in other reviews. In general, the information is well summarized and justifies its importance in the study of influenza viruses. However, I have some comments:

 1.     Throughout the document there are some inaccuracies or omissions that need to be corrected. Taxonomic categories (such as Orthomyxoviridae) should be written in italics. Where it is mentioned that there are four subtypes of influenza viruses, it is incorrect, they are four genera and within each genus there is only one species, called influenza A, B, C and D viruses (https://ictv.global/taxonomy). Subtypes refer to a subdivision within the influenza A virus species.

2.     Birds are not necessarily the natural hosts of all influenza A viruses, at least H17 and H18 viruses have been found only in bats (https://doi.org/10.1128/JVI.01357-19¸ https://doi.org/10.3390/ v13040547).

3.       Line 34: Check the writing of the figure (90,000-67,0000).

4.       I recommend using sialic acids instead of sialic acid, since it is a family of molecules..

5.       Line 67: I recommend modifying the wording, since they are not the two most common but two of the most common sialic acids. There are some other important types of sialic acid.

6.       Check that abbreviations such as CFG, CAMH gene, LacNA, Gal, GlcNAc, etc., are correctly written, in addition, in all cases add the meaning the first time they are used.

7.       Description of Figure 1: They are not different sialic acids, but different sialylated molecules or sialylated glycan molecules. The wording of this needs to be revised throughout the manuscript.

8.       Line 127: Should not be used “traditionally” but something like “commonly”. Something similar is what is written in line 409.

9.       The information in the paragraph between lines 126 and 140 is not referenced.

10.   Line 155: What is SA1?

11.   Line 224: The cited reference does not say “affinity maturation” but rather “avidity maturation”.

12.  Line 239: Check if the displayed structure of 3'SiaTF is correct or is another (Neu5Ac 2-3 Gal β1-3 GalNAc?)

Author Response

Thank you for the helpful comments. Based on the comments and suggestions, we have made careful modifications to the original manuscript. Details of our point-by-point responses are in the attachment, and all revisions made to the manuscript have been marked up using the“Track Changes” function.
